# Mesothelioma Mouse Models with Mixed Genomic States of Chromosome and Microsatellite Instability

**DOI:** 10.3390/cancers14133108

**Published:** 2022-06-24

**Authors:** Yurong Song, Shaneen S. Baxter, Lisheng Dai, Chelsea Sanders, Sandra Burkett, Ryan N. Baugher, Stephanie D. Mellott, Todd B. Young, Heidi E. Lawhorn, Simone Difilippantonio, Baktiar Karim, Yuwaraj Kadariya, Ligia A. Pinto, Joseph R. Testa, Robert H. Shoemaker

**Affiliations:** 1Cancer ImmunoPrevention Laboratory, Frederick National Laboratory for Cancer Research, Frederick, MD 21702, USA; shaneen.baxter@nih.gov (S.S.B.); lisheng.dai@nih.gov (L.D.); pintol@mail.nih.gov (L.A.P.); 2Animal Research Technical Support of Laboratory Animal Sciences Program, Frederick National Laboratory for Cancer Research, Frederick, MD 21702, USA; chelsea.sanders@nih.gov (C.S.); difilips@mail.nih.gov (S.D.); 3Mouse Cancer Genetics Program, National Cancer Institute, Frederick, MD 21702, USA; burketts@mail.nih.gov; 4CLIA Molecular Diagnostics Laboratory, Frederick National Laboratory for Cancer Research, Frederick, MD 21702, USA; ryan.baugher@nih.gov (R.N.B.); stephanie.mellott2@nih.gov (S.D.M.); todd.young@nih.gov (T.B.Y.); heidi.lawhorn@nih.gov (H.E.L.); 5Molecular Histopathology Laboratory, Frederick National Laboratory for Cancer Research, Frederick, MD 21702, USA; baktiar.karim@nih.gov; 6Cancer Signaling and Epigenetics Program, Fox Chase Cancer Center, Philadelphia, PA 19111, USA; yuwaraj.kadariya@fccc.edu (Y.K.); joseph.testa@fccc.edu (J.R.T.); 7Chemopreventive Agent Development Research Group, Division of Cancer Prevention, National Cancer Institute, Bethesda, MD 20892, USA; shoemakr@mail.nih.gov

**Keywords:** mesothelioma, microsatellite instability, chromosome instability, genomic instability, mouse model, cell line, immunotherapy, biomarker

## Abstract

**Simple Summary:**

Only a limited number of murine mesothelioma cell lines have been developed to date. We sought to expand this number and to characterize the models in detail to enable studying mesothelioma biology in vivo. Two cell lines were identified as showing well-defined mesothelioma biomarkers and being suitable for preclinical use. In the course of our studies, we observed a mixed phenotype of chromosomal instability and microsatellite instability not previously reported in mouse models. Moreover, microsatellite markers were detectable in the plasma of tumor-bearing animals, which potentially can be used as non-invasive biomarkers for early cancer detection and monitoring the effects of interventions.

**Abstract:**

Malignant mesothelioma (MMe) is a rare malignancy originating from the linings of the pleural, peritoneal and pericardial cavities. The best-defined risk factor is exposure to carcinogenic mineral fibers (e.g., asbestos). Genomic studies have revealed that the most frequent genetic lesions in human MMe are mutations in tumor suppressor genes. Several genetically engineered mouse models have been generated by introducing the same genetic lesions found in human MMe. However, most of these models require specialized breeding facilities and long-term exposure of mice to asbestos for MMe development. Thus, an alternative model with high tumor penetrance without asbestos is urgently needed. We characterized an orthotopic model using MMe cells derived from *Cdkn2a^+/−^;Nf2^+/−^* mice chronically injected with asbestos. These MMe cells were tumorigenic upon intraperitoneal injection. Moreover, MMe cells showed mixed chromosome and microsatellite instability, supporting the notion that genomic instability is relevant in MMe pathogenesis. In addition, microsatellite markers were detectable in the plasma of tumor-bearing mice, indicating a potential use for early cancer detection and monitoring the effects of interventions. This orthotopic model with rapid development of MMe without asbestos exposure represents genomic instability and specific molecular targets for therapeutic or preventive interventions to enable preclinical proof of concept for the intervention in an immunocompetent setting.

## 1. Introduction

Malignant mesothelioma (MMe) is a rare malignancy originating from the linings of the pleural, peritoneal and pericardial cavities. Most MMe arise from the pleura (malignant pleural mesothelioma, MPM), while peritoneal mesothelioma (PM) accounts for 7–30% of cases [1]. MPM predominates in men, whereas the prevalence for PM is the same for men and women in the U.S. The well-known risk factor for MMe is asbestos exposure (~80% for MPM and 33–50% for PM). Other risk factors include radiation, thorium, and other carcinogenic minerals, such as erionite and mica. It has been shown that MMe has a high frequency of *CDKN2A* deletions [2], inactivating mutations in *NF2* [3,4] and *BAP1* [5,6,7]. Recent next-generation sequencing (NGS) analyses of MPM have revealed significantly mutated *BAP1*, *NF2*, *TP53*, *SETD2*, *DDX3X*, *ULK2*, *RYR2*, *CFAP45*, *SETDB1* and *DDX51*, recurrent mutations in *SF3B1* (~2%) and *TRAF7* (~2%) [8], and a novel subtype with the overexpression of the immune checkpoint gene *VISTA* [9]. Integrated analysis shows that alterations in several signaling pathways (e.g., Hippo, mTOR, histone methylation, RNA helicase and p53) may drive MPM tumorigenesis [8]. 

Asbestos-exposed mice with heterozygous deletions of *Cdkn2a*, *Nf2*, and *Bap1* have each been shown to have an increased risk of MMe development compared to wildtype mice [10,11,12,13,14], supporting the importance of these tumor suppressor genes as drivers in MMe pathogenesis. Moreover, it has been shown that homozygous loss of *Cdkn2a* is a common driver of MMe tumorigenesis induced by asbestos in wildtype murine models [15]. 

The current treatment options for human MMe are systemic chemotherapy, molecular and immunotherapy, and surgery. Despite the reduction in and strict regulation of asbestos use and significant ongoing research to identify molecular drivers, the survival improvements for MMe over the recent decades have only been modest [16,17]. Early phase clinical trials of immune checkpoint inhibitors showed an overall median survival of 7–17 months in a small subset of pretreated patients, while most patients experienced treatment failure [18,19,20]. Nivolumab and ipilimumab is the first drug regimen approved by the FDA for pleural MMe since 2020, and a 4-month improvement in overall survival was observed in pleural MMe patients who received this drug combination compared to those receiving cisplatin or carboplatin plus pemetrexed [21]. However, only a minority of MMe patients respond to immune checkpoint inhibitors, and drug resistance develops in nearly all MMe cases. Thus, more studies are needed to better understand the biology of MMe for therapy selection and patient stratification, and suitable immunocompetent mouse models recapitulating human MMe are required for preclinical testing of therapeutic treatment and preventive strategies. 

Several genetically engineered mouse (GEM) models have been generated by introducing the same genetic lesions found in human pleural MMe in the mesothelial lining of the thoracic cavity of mice [22,23,24,25,26,27,28,29,30]. Mice with a heterozygous deletion of both *Cdkn2a* and *Nf2* (*Cdkn2a^+/−^;Nf2^+/−^*) developed MMe following chronic injection of 400 ug crocidolite asbestos every 3 weeks starting at 6–8 weeks of age [31]. Since these GEM models require specialized breeding facilities and injections of either adenovirus that expresses Cre recombinase or repeated injection of asbestos for MMe to develop with variable tumor incidence, an alternative model with 100% penetrance without the need for injections of adenovirus or asbestos is urgently needed for therapeutic drug screening and preventative vaccine development. 

Several murine MMe cell lines (e.g., AB1, AB12, AB22, AK7) have been generated from ascitic fluid or peritoneal lavage from spontaneously arising MMe tumors in wildtype mice exposed to asbestos [22,23,32] and used to establish syngeneic subcutaneous or orthotopic mouse models [26] for chemo- and immuno-based therapies [33]. Recently, murine MMe cell lines from *Cdkn2a^+/−^;Nf2^+/−^* mice exposed to asbestos have been developed with genetic alterations closely resembling human MMe [31]. We sought to develop orthotopic MMe mouse models using MMe cells derived from the asbestos-exposed *Cdkn2a^+/−^;Nf2^+/−^* model. Tumorigenicity studies indicate that these tumor cells grew very well in syngeneic mice via intraperitoneal injection without the need for further asbestos exposure. It has been reported that human MMe has a very low mutation burden [34] and only a few recurrently mutated driver genes (mainly tumor suppressor genes) [8,9]. Thus, genomic instability may be more relevant in MMe pathogenesis than nucleotide-level activating mutations [35]. To this end, we assessed the status of chromosome and microsatellite instability in MMe cell lines and orthotopic tumor model.

## 2. Materials and Methods

### 2.1. MMe Cells 

Six MMe cell lines were derived from ascitic fluid and/or peritoneal lavage of asbestos-injected, tumor-bearing *Cdkn2a^+/−^;Nf2^+/−^* mice in a FVB/NCrl background [31]. PCR-based pathogen testing showed that sixteen mouse pathogens were negative, including mycoplasma (data not shown). Cells were thawed and asynchronously grown in DMEM with L-Glutamine, 10% FBS, and 1% penicillin-streptomycin at 37 °C with 5% CO_2_. Cell viability was assessed using trypan blue and AutoCell 2000 Cellometer (Nexcelom Biosciences, LLC, Lawrence, MA, USA). Cell morphology was assessed using an inverted Leica microscope (Leica Biosystems, Wetzlar, Germany) and images were taken using an EVOS FL Cell Imaging System (Thermo Fisher Scientific, Waltham, MA, USA). Genetic profiling via Short Tandem Repeats (STR) was conducted using a multiplex PCR-based assay to establish a reference profile using a panel of 9 microsatellite markers (CellCheck^TM^ Mouse, IDEXX BioAnalytics, Columbia, MO, USA). MHC class I haplotypes were assessed by flow cytometry analysis (FACSCELESTA HTS, BD Biosciences, Franklin Lakes, NJ, USA). 

For tumorigenicity studies, MMe cells grown in T-75 or T-150 flasks were washed with PBS twice, and then trypsinized with 0.05% trypsin-EDTA solution. After 5 min incubation, culture media were added to the flasks. Cells were pelleted and then resuspended in serum free medium (SFM) for cell counting. Different cell inocula were injected subcutaneously (s.c.) at 100 uL or intraperitoneally (i.p.) at 100 or 500 uL into 7–8-week-old syngeneic mice. 

### 2.2. Animals

FVB/NCrl females or males (Charles River) at 7–8 weeks of age were used as recipients in tumorigenicity studies. Animals were randomized into study groups based on their age and body weight. Following inoculation of MMe cells s.c., animals were palpated twice per week. Tumor volume (TV) was measured twice per week using calipers and calculated using the formula *length × width × height × Pi/6*. Take rate was analyzed based on whether the animal had a measurable tumor. Body weight (BW) was measured once weekly. Animals inoculated i.p. were monitored daily for signs of distress once they demonstrated a 10% increase in body weight attributable to ascites. Abdomen circumference and body weight were measured once weekly before ascites developed and twice a week after ascites developed. The animals were euthanized once they showed signs of illness or s.c. tumors reached 20 mm maximal allowable size based on NCI-Frederick ACUC guidelines. Full necropsy was performed to assess the metastases and overall tumor burden. For animals with i.p. inoculation, all internal organs were assessed, harvested, and weighed. Terminal blood was collected via cardiac puncture for plasma, and tumor tissues were harvested for fixation in 10% NBF for histopathological evaluation and marker analysis.

All animals in this study were monitored daily and wet food was provided when they showed signs of illness. All mice were euthanized by CO_2_ asphyxiation per NCI-Frederick ACUC guidelines to minimize pain and suffering. NCI-Frederick is accredited by AAALAC International and adheres to the Public Health Service Policy for the Care and Use of Laboratory Animals and the procedures outlined in the “Guide for Care and Use of Laboratory Animals” (National Research Council; 1996; National Academy Press; Washington, DC, USA).

### 2.3. Histology and Immunohistochemistry

Cell pellets or tumor tissues were fixed in 10% neutral-buffered formalin (NBF), and formalin-fixed paraffin-embedded (FFPE) blocks were made and sectioned for Hematoxylin and Eosin (H&E) and immunohistochemistry (IHC) staining, as described previously [36]. The antibodies used are listed in Appendix A. 

### 2.4. Cell DNA and Plasma cfDNA Extraction

DNA was isolated from cultured cells using the DNeasy^®^ Blood & Tissue Kit from Qiagen according to the manufacturer’s recommendations. Cell-free DNA (cfDNA) was isolated from mouse plasma using the Quick-cfDNA/cfRNA™ Serum & Plasma Kit (Zymo Research, Irvine, CA, USA; Cat. No. R1072) as recommended by the manufacturer. DNA quantification was carried out using Qubit^®^ dsDNA HS Assay kit (Cat No. Q33231) and the Qubit™ Flex Fluorometer (Cat. No. Q33327) from Thermo Fisher Scientific (Waltham, MA, USA), according to the manufacturer’s instructions.

### 2.5. Fragment Analysis and Sanger Sequencing to Assess Microsatellite Instability (MSI)

Primers were designed for mouse microsatellite loci L24372-A27, U12235-A24, mBat30, mBat37, mBat64, and mBat67 on mouse build GRCm38.p6 [37,38,39,40,41,42]. All primers were ordered as stated in Appendix A (Integrated DNA Technologies, Coralville, IA, USA) and amplified with Platinum™ SuperFi™ PCR Master Mix (12.5 µL; Thermo Fisher Scientific, Waltham, MA, USA) with kit-provided GC Enhancer (5 µL), with the exception of mBat67, where molecular grade water (5 µL) was substituted in place of GC Enhancer. Sample input was 5 µL of 0.5–20 ng/µL DNA, using 1.25 µL each of 10 µM primer. Amplification was performed on ProFlex PCR System (Thermo Fisher Scientific) using PCR conditions, as stated in Appendix A. The resulting products were then checked for quality and concentration with 2100 Bioanalyzer and DNA 1000 kit (Agilent Technologies, Santa Clara, CA, USA). Samples amplified with Fragment Analysis primers were prepared for running on fragment analysis by diluting with water (up to 1:15 ratio). A master mix was created using 1 µL of diluted sample, Hi-Di™ Formamide (8.5 µL per reaction; Thermo Fisher Scientific), and GeneScan™ 500 LIZ™ dye size standard (0.5 µL per reaction; Thermo Fisher Scientific), and incubated with the ProFlex™ PCR System. Samples were then processed on 3730xl DNA Analyzer (Thermo Fisher Scientific) using 96 capillary 50cm array and DS-33 Matrix Standard Kit (Dye Set 5) and 3730XL Data Collection Software (version 5.0; Thermo Fisher Scientific). Data were then analyzed and overlayed using GeneMapper™ software (version 5.0; Thermo Fisher Scientific). Instability of the examined locus in a sample was defined by shifted peaks or altered length of the PCR product compared to a wt control. Instability at 2 or more microsatellite loci was defined as MSI-High (MSI-H) and instability at 1 locus or none as microsatellite stable (MSS).

For mBat67, confirmatory Sanger sequencing was performed. Samples were diluted and purified using exonuclease I (GE Healthcare, Pittsburgh, PA, USA) and shrimp alkaline phosphatase (SAP; Affymetrix USB) by following the Exo-Sap protocol. The incubation was carried out in the ProFlex™ PCR System: 37 °C for 15 min, then 80 °C for 15 min, followed by a 4 °C hold. This purified amplicon then proceeded into cycle sequencing with BigDye™ Terminator v3.1 Cycle Sequencing Kit (Thermo Fisher Scientific) and M13 Forward and M13 Reverse primers (Invitrogen, Waltham, MA, USA), using the following conditions in the ProFlex™ PCR System: 96 °C for 1 min, 25 cycles of 96 °C for 10 s, 50 °C for 5 s, and 60 °C for 1 min and 15 s, followed by a hold at 4 °C. Samples were then processed on an 3730xl DNA Analyzer using 96 capillary 50 cm array, 3730/3730XL DNA Analyzer Sequencing Standards, BigDye™ Terminator v3.1 Kit and 3730XL Data Collection Software (version 5.0; Thermo Fisher Scientific). Data were then analyzed using Mutation Surveyor (version 5.1.2; SoftGenetics, State College, PA, USA).

### 2.6. Spectral Karyotyping (SKY) Analysis to Assess Chromosome Instability (CIN)

The metaphases of cultured cells were arrested for three hours prior to harvest using Colcemid (10 ug/mL; 15210-040, KaryoMAX ^®^ Colcemid Solution, Invitrogen, Carlsbad, CA, USA), and then treated with hypotonic solution (KCl 0.075M, MK-6858-04, Macron Chemicals, Capitol Scientific, Austin, TX, USA) for 15 min at 37 °C and fixed with methanol: acetic acid (3:1). Slides were prepared and aged overnight for SKY analysis. The metaphases were hybridized with the 21-color mouse SKY paint kit (FPRPR0030, Applied Spectral Imaging (ASI), Carlsbad, CA, USA) in a humidity chamber at 37 °C for 16 h according to the manufacturer’s protocol [43], and then washed with 0.4xSSC at 72 °C for 4 min. Spectral images of the hybridized metaphases were acquired using Hyper Spectral Imaging System (ASI, Carlsbad, CA, USA) mounted on top of an epi-fluorescence microscope (Imager Z2, Zeiss, Thornwood, NY, USA), and analyzed using HiSKY 7.2 acquisition software (GenASIs, ASI, Carlsbad, CA, USA). G-banding was simulated by the electronic inversion of DAPI counterstaining. An average of 10–15 mitoses of comparable staining intensity and quality were examined and compared per cell line for chromosomal abnormalities. The standard ideogram of banding patterns for mouse chromosomes was compared to determine the karyotype in each cell [44].

### 2.7. Statistical Analysis

Log-rank (Mantel-Cox) test of Kaplan–Meier survival curves was performed for survival analysis using GraphPad Prism 9. Two-tailed *t*-test and Analysis of Covariance (ANOVA) adjusted for group and body weight using SAS were performed to evaluate the statistical significance of mean circumference between MMe cell-injected and age-matched non-cell-injected control groups. *p* < 0.05 was considered statistically significant.

## 3. Results

### 3.1. Cell Line Characterization

All six MMe cell lines had very high viability (>90%), and morphologically looked like epithelial cells (Appendix A). Genetic profiling of all cell lines using STR markers matched the reference marker profile of FVB/NCrl mice, which confirmed the genetic background of these cell lines (Appendix A). Thus, FVB/NCrl mice were used as recipients for the subsequent tumorigenicity studies described below. 

MHC class I molecules are critical components of antigen presentation to cytotoxic T cells. It is well known that tumor cells can escape immune surveillance by downregulating or mutating MHC I molecules or β2-microglobulin (B2M). To determine if MMe cells expressed the MHC I haplotypes, we assessed the expression of H-2Kq, H-2Dq/H-2Lq by flow cytometry analysis. We found that all six cell lines had very high expression of all three haplotypes (Appendix A), indicating that antigen presentation in these tumor cells may be not compromised. 

Mesothelin (MSLN) is usually expressed at low levels in mesothelial cells lining the pleura, peritoneum, and pericardium. However, it is highly expressed in MMe. Thus, it has been used as one of the markers for this disease. We assessed MSLN expression by flow cytometry analysis and IHC staining using FFPE blocks prepared from cell pellets. As expected, MSLN was highly expressed in all six cell lines, although at variable levels (Appendix A). MM96 had the highest expression (94.1%) by flow analysis. IHC staining showed strong membrane and cytoplasmic expression of MSLN in all six cell lines (Appendix A and Appendix A). To confirm that the cells established in culture were indeed MMe cells, we further analyzed an MMe-specific marker Wilms Tumor Protein 1 (WT1) by IHC. It was strongly expressed in all six cell lines (Appendix A), indicating they are mesothelial in origin.

### 3.2. MMe Cells Are Tumorigenic In Vivo following i.p. Injection

To determine if MMe cells can grow in syngeneic mice, tumorigenicity studies were carried out by i.p. injection of different inoculum of MMe cells into 7–8-week-old FVB/NCrl mice (*n* = 10 per group). All six MMe cells were tumorigenic (Table 1). However, the take rate was variable. Animals inoculated with MM201 (2.5–5 × 10^6^), MM96 (5 × 10^6^), MM87 (1 × 10^7^), or MM58 (5 × 10^6^) cells had a 100% take rate, but not animals with MM410 (5 × 10^6^) and MM380 (1 × 10^7^). The median survival was much shorter for animals injected with MM58 or MM87 (23 and 27 days post inoculation (dpi), respectively) compared to animals with MM201 and MM96 (68 and 101 dpi, respectively) (Table 1 and Figure 1 bottom panel), demonstrating the aggressiveness of MM58 and MM87 cells. This was further evidenced by the necropsy and histology findings that all recipient animals injected with MM58 or MM87 developed ascites and had jaundice and/or massive liver necrosis except one animal that we were unable to assess (found dead; data not shown). Compared to age-matched mice not injected with MMe cells, animals inoculated with MM96 or MM201 cells had decreased body weight (Figure 1, middle panel) but increased mean circumference for MM96 cells (Figure 1, top left panel) or no change in circumference for MM201 cells (Figure 1, top right panel). MMe cells were originally derived from males. To determine if there was a recipient gender effect on tumor growth, we tested MM201 cells in both male and female recipients. They all had a 100% take rate with median survivals of 68–70 dpi, which was the same as that in female recipients (68 dpi) (Table 1). Histology analysis showed no difference in tumors from male and female recipients (data not shown). 

Full necropsy was performed on all MMe-inoculated animals. Ascites was observed in some but not all animals. Tumor cells grew as nodules in the peritoneum or along the serosal surfaces (e.g., diaphragm, omentum, pancreas, ovary, kidney, and liver) and in mesenteric adipose tissue, and developed metastases such as in liver and pancreas (Figure 2a). Histologic analysis revealed poorly differentiated tumors and invasion into the serosal layer of intestine and stomach (Figure 2a), which recapitulate the de novo tumors in the *Cdkn2a^+/−^;Nf2^+/−^* mouse model from which these MMe cell lines were derived (Appendix A). Necrosis in liver were also observed in orthotopic models as that in *Cdkn2a^+/−^;Nf2^+/−^* model (Appendix A). We further characterized MMe tumors by IHC on FFPE sections (Appendix A and Figure 2). They were highly proliferative based on Ki67 and strongly positive for MSLN and WT1 (Figure 2b). TERT is usually overexpressed in up to 90% of human primary cancers [45,46]. We assessed TERT expression by IHC and found that MMe tumors had high expression of TERT (Figure 2b). Based on in vivo tumor growth and marker expression, MM201 and MM96 are suitable for use to establish orthotopic models for testing preventive and therapeutic interventions. 

It is evident that tumor microenvironment plays a critical role in tumor growth in vivo [47,48,49,50,51,52,53]. To determine if these MMe cells could grow in an s.c. condition, we next assessed tumorigenicity via s.c. injection of various inocula of MM96 into one flank of 7–8-week-old FVB/NCrl mice (*n* = 10 per group) (Table 1). We found that animals inoculated at this ectopic site with up to 5 × 10^6^ cells did not result in a 100% take rate after 114 dpi (Table 1). To determine if the tumors grown in the s.c. setting still maintained the expression of MSLN and WT1, we assessed their expression on FFPE sections via IHC and found that MSLN was expressed at variable levels and WT1 was strongly positive (Appendix A). To determine if cell proliferation could account for slow tumor growth and a low take rate, we performed Ki67 IHC. In contrast to i.p. tumors formed by these MMe cells, we found that MM96 s.c. tumors had very low proliferation rate (Appendix A), which explains the slow tumor growth and low take rate in the s.c. setting. This indicates that growth in the native microenvironment is preferred by these MMe cells.

### 3.3. Chromosomal Instability (CIN) in MMe Cells

Whole-exome sequencing (WES) studies have shown that human pleural MMe have a very low average number of somatic mutations. In one investigation, the number of mutations per tumor ranged from 2 to 52, corresponding to an average of 0.79 mutations per megabase (range: 0.07–1.71) [34]. A larger WES study, performed by The Cancer Genome Atlas (TCGA) revealed low somatic mutation rate (<2 non-synonymous mutations per megabase) in all samples except one (eight non-synonymous mutations per megabase) [9]. Thus, it has been proposed that CIN may be pathogenic for MMe development [35]. To this end, we analyzed chromosome changes at a macrolevel via spectral karyotyping (SKY) in MMe cells. We found significant chromosomal abnormalities in all six MMe cell lines (Figure 3 and Appendix A). Polysomy was readily detected, as shown by the presence of extra copies of one or more whole chromosomes. Translocations were also observed in all six MMe cell lines, such as T(1;15) in MM87, unbalanced derivative translocations Der(X)T(X;9) in MM410, and Der(4)T(4;8) in MM201 (Figure 3). Translocations frequently occurred on chromosomes 3, 4, 5, 8, 12, and 13 (Appendix A). Interestingly, we observed heterogeneity of CIN in these MMe cells. Using chromosome X- and Y-specific probes, we confirmed X and Y chromosome abnormalities in MMe cells (Appendix A). X and Y chromosome translocations and two to three copies of X or Y chromosome were observed. 

### 3.4. Microsatellite Instability (MSI) in MMe Cells and Plasma from Tumor-Bearing Animals

A recent comprehensive study showed that MSI was present in 2.4% of human mesothelioma [54]. This prompted us to assess the MSI status in murine MM201, MM87, and MM410 cells using six MSI markers (mBat30, mBat37, mL24372, mU12235, mBat64, and mBat67) and fragment analysis. After PCR amplification using fluorescence-labeled primers, fragments generated from each MMe cell line and wildtype (wt) control along with size standard were combined and separated by size using capillary electrophoresis (CE). 

Three different fluorescent dyes were detected in each sample, such as MMe cells as blue, wt as green, and size standard as orange. We observed that microsatellite peaks of two markers (mBat64 and mBat67) in MMe cells (blue) shifted to the left of wt (green), indicating a smaller fragment size and deletions of As in MNRs in MMe cells (Figure 4 and Figure 5 and Appendix A), while peaks of the other four markers in MMe cells overlapped with the peaks in wt, indicating no deletions in these MNRs in MMe cells. The MSI profile was consistent among the three MMe cell lines we assessed. The mBat64 locus showed a deletion of 43 As compared to wt, while the mBat67 locus only had one big peak upstream of the wt position (Figure 5a), indicating a large deletion. Sanger sequencing confirmed that these cells indeed had a deletion of 258 bp around the MNR (Figure 5b and Appendix A; GGGGGCTGGTGAGATGGCTCAGTGGGTAAGAGCACCCGACTGCTCTTCCGAAGGTCCAAAGTTCAAATCCCAGCAACCACATGGTGGCTCACAACCACCCGTAATGAGATCTGATGCCCTCTTCTGGAGTGTCTGAAGACAGCTACAGTGTACTTACATATAATAAATAAATAAATCTTTAAAAAAAAAAAAAAAAAAAAAAAAAAAAAAAAAAAAAAAAAAAAAAAAAAAAAAAAAAAAAAAAAAAACAAAATATTCT). To determine if the MSI profile in MMe cells was expressed during in vivo tumorigenesis in a way that could be useful as a biomarker of tumor presence or burden, we assessed plasma cfDNA from animals inoculated with MM58, MM410, and MM201. Indeed, the same MSI profile was detected in plasma as that in MMe cells (Appendix A), confirming that the MMe tumors had mixed genomic states characterized by CIN and MSI and that plasma cfDNA can be monitored as an indicator of mesothelioma tumor burden. 

## 4. Discussion

Malignant mesothelioma, a uniformly lethal cancer with poor response to current therapies, has been extensively studied over the years. The results from current therapeutic studies including immunotherapies have been generally disappointing in pleural MMe [20,55,56,57,58], although the combination of nivolumab and ipilimumab did result in a 4-month improvement in overall survival in patients who received this drug combination compared to those receiving a platinum drug plus pemetrexed [21]. However, only a minority of pleural MMe patients respond to immune checkpoint inhibitors, so more effective therapies are urgently needed. To facilitate the understanding of the biology of MMe and to test new therapies, several mouse models have been developed by introducing the same genetic lesions found in human MMe [25]. Although these models recapitulate the human disease including extensive inflammatory responses, the complex breeding to engineer several genetic lesions and the need for either an injection of adeno-Cre virus in conditional knockout strains or asbestos in heterozygous mice make these models less accessible to the general research community for preclinical therapeutic evaluation. 

We developed an orthotopic MMe model with i.p. injection of MMe cells derived from ascitic fluid and/or peritoneal lavage from *Cdkn2a^+/−^;Nf2^+/−^* mice exposed to asbestos. Six MMe cell lines were characterized in vitro and in vivo via i.p. injection. We found that all six MMe cells were tumorigenic in vivo. Although there was no difference on in vitro growth among six cell lines, we found that MM87 and MM58 grew aggressively, with median survival of less than a month, while MM380 and MM410 did not have a 100% take rate. Several features make MM201 and MM96 suitable for preclinical testing: 1. good tumor growth in immune competent mice with a 100% take rate and relative short median survival (2–3 months) compared to GEM mice; 2. tumors harbor the same genetic lesions and were induced by exposure to carcinogenic asbestos, as in the human disease counterpart; 3. high expression of TERT, MSLN, and WT1; 4. ease of setting up a large study cohort without the need for subsequent injections of adenovirus or asbestos. It is noteworthy that these MMe cells did not grow well in a s.c. setting, suggesting that microenvironment is critical for tumor growth as reported in numerous reports in the literature [48,49,50,51].

Genomic instability, one of the hallmarks of cancer cells [59], can be manifested through CpG island methylator phenotype (CIMP) variations at the nucleotide level such as base pair mutation and MSI, or variations at the chromosome level (CIN) [60,61,62]. It is generally thought that defects in DNA repair mechanisms lead to exclusive states of CIN and MSI [63]. NGS studies have shown that MMe has a very low mutation burden [9,34]. Thus, it has been proposed that CIN may be pathogenic in MMe development [35]. By SKY analysis, we found that all six MMe cell lines derived from *Cdkn2a^+/−^;Nf2^+/−^* mice showed polysomy and chromosome translocations with intratumoral heterogeneity, suggesting chromosome instability in these MMe cells. Sneddon et al. showed copy number changes in murine MMe cell lines derived from asbestos-exposed wildtype murine models of MMe via whole-exome sequencing [15]. More recently, Wahlbuhl et al. reported chromosomal aberrations and cytogenetic changes in three murine MMe cell lines AB1, AB22, and AC29 derived from female Balb/c and CBA mice inoculated with asbestos fibers [32]. Consistently, a high number of micronuclei containing whole chromosomes or damaged chromosome fragments was observed in peripheral blood lymphocytes in human pleural MMe, although there was no association between micronuclei and asbestos exposure [64]. Moreover, multiple chromosomal aberrations and copy number variations have been reported in human MMe [65,66,67,68,69,70,71,72,73]. The CIN phenotype is also found in other cancer types [74]. In addition, it has been shown that CIN-mediated intratumoral heterogeneity correlates with an increased risk of death or recurrence [75], indicating a potential prognostic value of CIN [76]. It has been reported that CIN can result from a series of genetic changes such as *KRAS*, *TP53*, *DCC/SMAD4*, and *APC* [77,78], or mutations/changes in MSH3 function [79,80]. Further investigation is needed to uncover the specific mutations in murine MMe cell lines derived from *Cdkn2a^+/−^;Nf2^+/−^* model and determine whether these mutations contribute to the CIN phenotype observed in these cells.

MSI is characterized by the accumulation of mutations (deletions/insertions of nucleotides) in microsatellite repeats (also known as short tandem repeats). It has been well studied in colorectal, endometrial, and gastric adenocarcinomas. A recent comprehensive sequencing study testing 2530 microsatellite loci with new analysis tools revealed that MSI was present in 27 different tumor types with variable disease-specific prevalence from 31.4% in endometrial carcinoma to 0.25% in glioblastoma multiforme, including cancer types in which MSI had not previously been well described, adrenocortical carcinoma (4.3%), cervical cancer (2.6%) and MMe (2.4%) [54]. The detection of MSI-H in human MMe prompted us to assess MSI status in murine MMe cells. Two of six MSI loci we tested showed deletions, indicating MSI in *Cdkn2a^+/−^;Nf2^+/−^* MMe cells. This is consistent with the notion that MSI may be under-detected in most cancers due to conventional microsatellite loci and detection technologies used [81,82]. These murine MMe cell lines genomically recapitulate a subset of human MMe tumors. Moreover, MSI high (MSH-H) MMe had, on average, a nearly 7-fold increase in mutational burden compared to microsatellite stable (MSS) MMe [54], suggesting that routine MSI screening for MMe patients should be considered to guide treatment decisions and stratification of MM patients for immunotherapy.

MMe cells derived from *Cdkn2a^+/−^;Nf2^+/−^* mice showed not only CIN, but also MSI. Traditionally, MSI and CIN are considered to be mutually exclusive pathways giving rise to sporadic cancers [83]. However, recent studies revealed mixed genomic instability states of CIN and MSI in colon cancer [84,85,86,87,88,89,90]. Shin et al. reported that the CIN phenotype was unexpectedly common in MSI-H colorectal cancer (CRC) tumors, and profiling diverse sequence tandem repeats in CRC revealed the co-occurrence of microsatellite and chromosomal instability involving increased copy number of chromosome 8 [80]. With the advancement of MSI detection technologies and new loci identified for MSI [54,82,91,92,93,94], it is highly likely that the mixed state of CIN and MSI will be discovered in cancers other than CRC, and both CIN and MSI can co-evolve to drive cancer progression and/or to be the consequence of cancer evolution during the progression of intratumorally heterogeneous genomic states [95].

Immunotherapy has been approved for many cancer types with high mutation burdens and MSI-H cancers (e.g., melanoma, colorectal cancer, and lung cancer) and is being actively pursued for the treatment of MMe. Analyses of different treatments from different clinical trials for MMe (e.g., immunotherapy versus platinum-based chemotherapy) have been performed and reported recently. Messori et al. found that nivolumab plus ipilimumab or pembrolizumab monotherapy showed a small but significant survival benefit, but not durvalumab plus pemetrexed plus cisplatin, compared to standard of care treatment (pemetrexed plus cisplatin) [96]. However, Meirson et al. [97] and Kerrigan et al. [98] reported no survival benefit for nivolumab plus ipilimumab or either a single agent alone compared to bevacizumab plus cisplatin plus pemetrexed or single-agent chemotherapy. MMe patients in these trials were not stratified by their MSI status or tumor mutation burden. Thus, it remains to be determined whether MMe patients with MSI-H would respond better to immunotherapy than MSS MMe patients. Moreover, immune response can be hampered by a complex network and many factors other than CIN and MSI. It has been shown that the expression of surface molecules (e.g., MHC class II, ICAM-1 and B7-2) in MMe cells is required for antigen presentation [99], and IFN-γ production in an anti-PPD (purified protein derivative) CD4^+^ T-cell clone was inhibited by TGF-β released by PPD-presenting MMe cells [100]. Thus, appropriate mouse models are needed for testing the effects of these factors on immune response to different therapies. The model we established here will be suitable to address whether the mixed MSI and CIN state will alter the response to immunotherapy.

It is noteworthy that among the six MSI mononucleotide loci we have tested so far, only two with long repeats (mBat64 and mBat67) showed a deletion, but not the other four with relatively short repeats (U12235, L24372, mBat30 and mBat37). This MSI profile is different from that in MSI-H CRC, which may indicate different MSI etiologies between two cancer types. Generally, MSI is considered a hallmark of mismatch repair deficiency (MMRD) due to mutated genes in Hereditary Non-Polyposis Colorectal Cancer (HNPCC) or Lynch Syndrome (*MLH1*, *MSH2*, *MSH6*, *PMS2*, and *EPCAM*). However, a recent study showed a novel association between the loss of DNA polymerase proofreading and MSI, which exerts distinct MSI signatures [101]. Moreover, childhood constitutional MMRD (CMMRD) cancers showed different mutated microsatellite loci and a lack of frequently mutated loci compared to adult MMRD cancers, indicating that MSI etiology may determine microsatellite loci and MSI profile in different cancers. Indeed, MSI at tetranucleotide repeats (elevated microsatellite alterations at selected tetranucleotide repeats; EMAST) was associated with mislocalized MSH3 from its normal nuclear location into the cytosol [102]. Furthermore, Shin et al. showed that the frameshift at microsatellite locus of *MSH3* exon 7 and the degree of EMAST were associated with the mixed CIN and MSI in CRC [80]. EMAST has been observed in multiple cancers [103,104] and a mouse model of colon cancer [105]. It is reasonable to speculate that different etiologies of MSI may lead to different responses to immunotherapy. It is not known if MMe tumor cells from *Cdkn2a^+/−^;Nf2^+/−^* mice have any replication repair deficiency. Further studies are needed to determine whether MSI in MMe cells is a result of an alternative mechanism other than MMR deficiency and whether MSI or mixed MSI and CIN states can be prognostic or predictive of immunotherapeutic response. 

The detectability of tumor MSI markers in plasma of tumor-bearing mice opens new avenues for experimental design where a non-invasive biomarker can be used to monitor the effects of preclinical interventions. If this phenomenon is also displayed in clinical mesothelioma with MSI, it may be useful for early cancer detection as well as monitoring the effects of interventions.

The model characterized in this report has several advantages. The MMe cell lines were derived from asbestos-induced peritoneal mesotheliomas in genetically engineered mice with deletions of two of the key tumor suppressor genes that have been implicated in the human disease counterpart. Thus, this model is suitable for studying mesothelioma biology in detail in vivo. Moreover, the model is also intended to represent genomic instability and specific molecular targets (e.g., Nf2/hippo, p16Ink4a/CDK4/6/Rb, and DNA repair pathways, as well as high expression of TERT and MSLN) for therapeutic or preventive interventions to enable a preclinical proof of concept for the intervention. The development of this type of information can be used to build a case for the clinical testing of the intervention. In addition, this model can be used in other advanced preclinical testing, such as toxicology testing, which is still necessary to support approval of a clinical protocol. However, the model also has limitations and challenges. For example, while our long-term goal for this model system is to use it to predict the therapeutic response to novel combined immunotherapies and/or new therapies for this uniformly deadly malignancy, as is well documented in the literature, therapeutic activity in a mouse model is not necessarily predictive of a human response. Neither our model system nor any other preclinical model would be expected to have absolute predictive value. The lack of activity could be the basis for a no-go decision on an intervention. Secondly, our model recapitulates only a subset of MMe since only *Cdkn2a* and *Nf2* were manipulated in this model. Other genes are also significantly mutated and play critical roles in MMe tumorigenesis (e.g., *BAP1*, *TP53*, and *SETD2*). Lastly, it is challenging to monitor a response to an intervention in an i.p. tumor model. The MMe cell lines can be engineered with luciferase expression for in vivo bioluminescence imaging. However, luciferase itself can be considered as a foreign antigen eliciting undesired immune response, which can be critical depending on the intervention.

## 5. Conclusions

We have successfully established a syngeneic orthotopic model using MMe cells derived from an asbestos-induced *Cdkn2a^+/−^;**Nf2^+/−^* MMe mouse model. The MMe cells showed a mixed state of CIN and MSI. Immunotherapy has been approved for pleural MMe. However, most patients experienced treatment failure. The model described in this study represents genomic instability and specific molecular targets for therapeutic or preventive interventions to enable a preclinical proof of concept for the intervention. It is easy to set up a study cohort without the need for an injection of adenovirus or asbestos. Further mechanistic analyses will potentially explain CIN and the different loci and scope of MSI that were introduced during tumorigenesis, especially by asbestos exposure in the *Cdkn2a^+/−^;Nf2^+/−^* model, which may lead to advancements in the treatment and management of MMe.

## Figures and Tables

**Figure 1 cancers-14-03108-f001:**
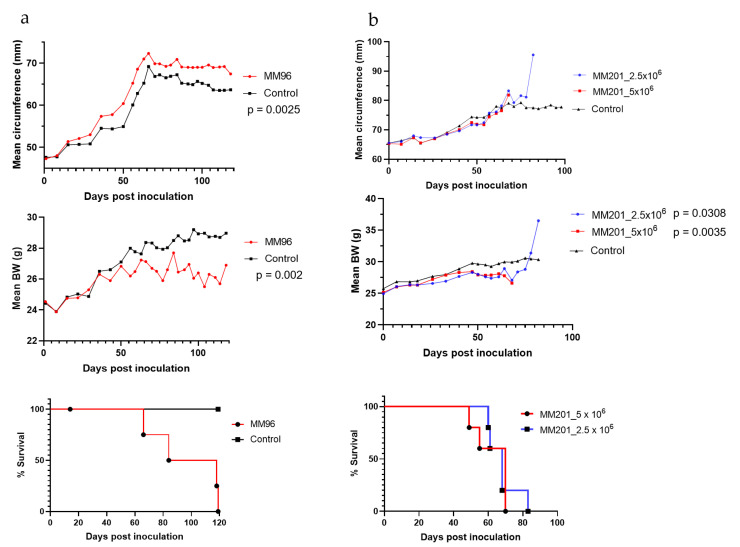
Mean circumference (top panel), mean body weight (middle panel) and Kaplan–Meier curve (bottom panel) in animals inoculated i.p. with (**a**) MM96 or (**b**) MM201 cells. Animals inoculated with SFM were used as control. One of the animals in MM96-injected group was found dead at 14 dpi due to injection related injury.

**Figure 2 cancers-14-03108-f002:**
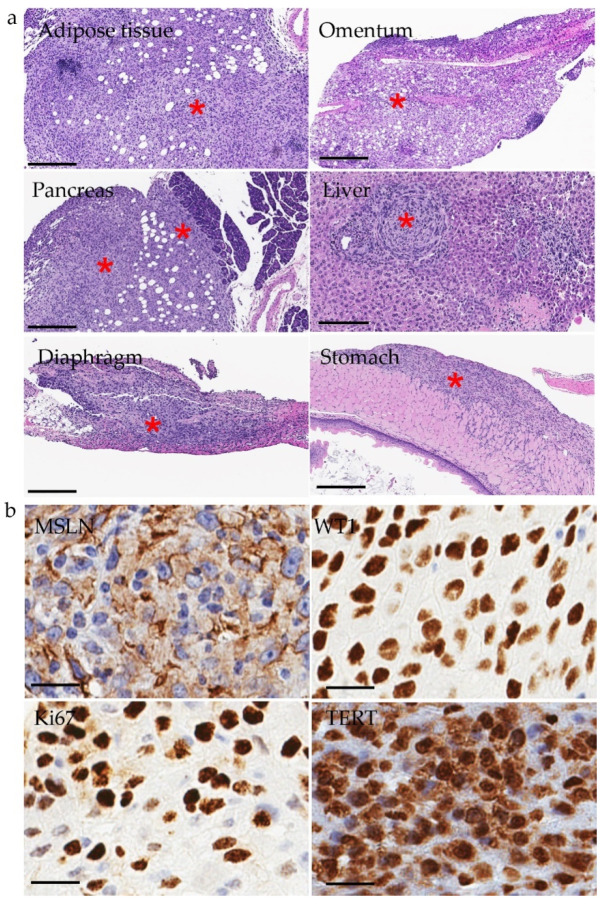
(**a**) H.E. staining of MM201 cells growing in mesenteric adipose tissue and along the serosal surfaces (e.g., omentum, pancreas, liver, diaphragm, and stomach). * Tumor cells. Scale bar: 200 um. (**b**) IHC staining of MSLN, WT1, Ki67, and TERT in MM201 i.p. tumors. Scale bar: 50 um.

**Figure 3 cancers-14-03108-f003:**
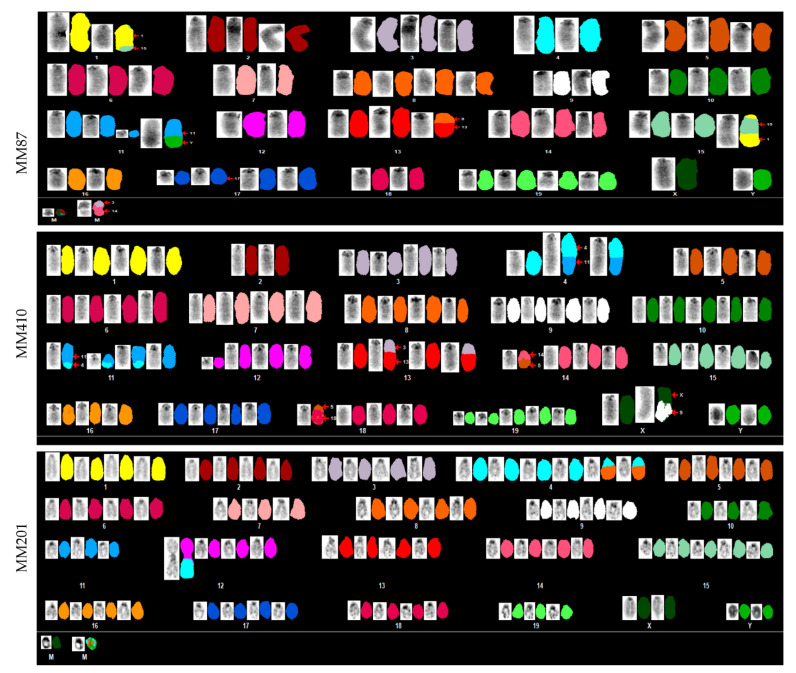
SKY analysis using probes labeled with 21-color mouse SKY paint for each chromosome in MMe cell lines. Representative karyotyping from MM87, MM410, and MM201 cell lines are shown here. Red arrows show examples of translocation. M, marker chromosome too little material for definite characterization.

**Figure 4 cancers-14-03108-f004:**
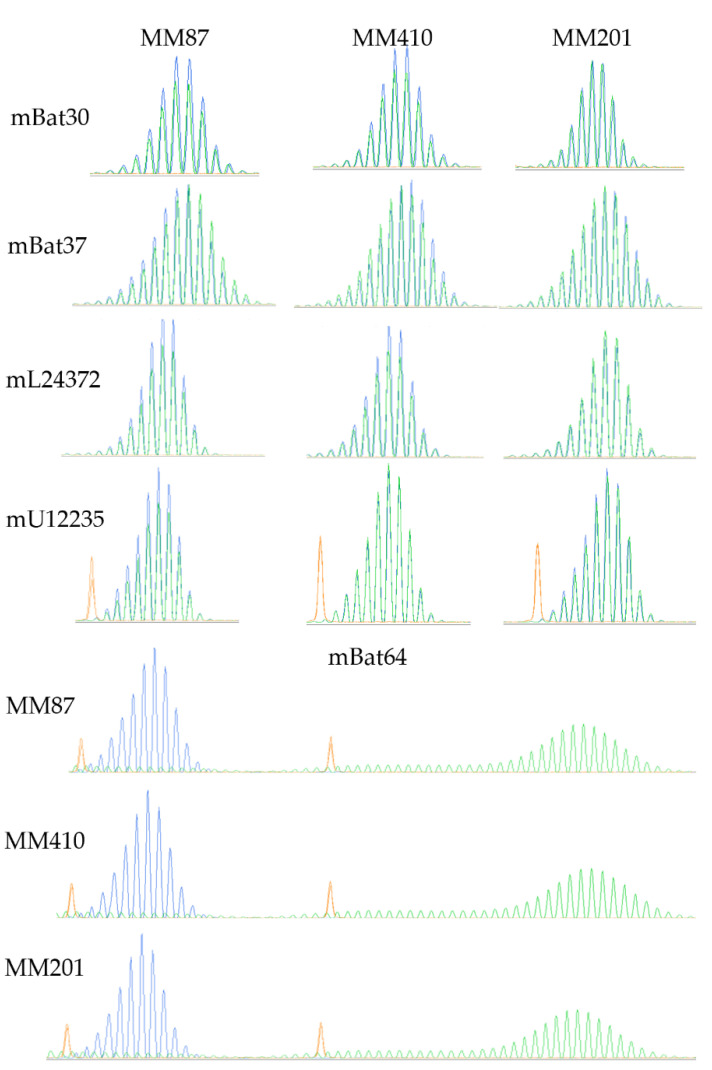
MSI detection in MMe cell lines by fragment analysis. MMe cells are in blue, wt tail DNA was used as control in green, and size standard is in orange.

**Figure 5 cancers-14-03108-f005:**
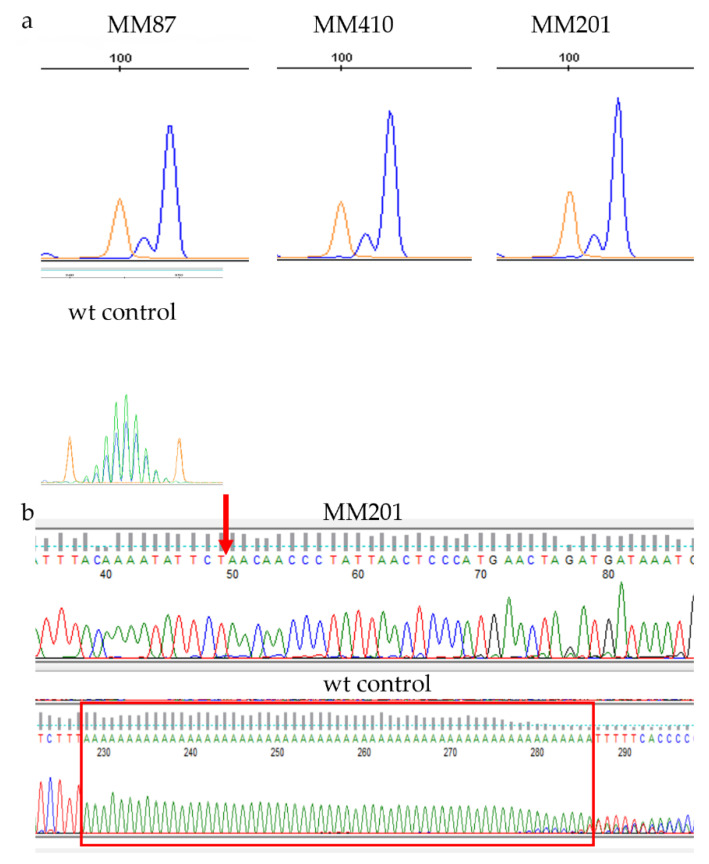
(**a**) MSI detection of mBat67 in MMe cell lines via fragment analysis. MMe cells are in blue, wt tail DNA was used as control in green, and size standard is in orange. wt control: wt mouse intestinal epithelial cells in blue. Peaks generated by MMe cells (blue) were upstream of peaks by wt tail (green; not shown). (**b**) Sanger sequencing confirmed a large deletion of 258 bp around the mBat67 locus. Red arrow indicates the deletion break point. wt control: wt mouse intestinal epithelial cells. Red box indicates mBat67 sequence detected in wt. Due to large amounts of slip-specific deletions, only 58 As sequence was detected by Sanger sequencing in wt control, which was used as wt mBat67 profile.

**Table 1 cancers-14-03108-t001:** Summary of tumorigenicity studies via i.p. and s.c. injection.

Cell Line	Inoculum (Volume as mL)	Take Rate (%)	Median Survival (dpi)
MM201	5 × 10^6^ (0.5)	100	68
	2.5 × 10^6^ (0.5) *	100	68
	5 × 10^6^ (0.5) *	100	70
MM96	5 × 10^6^ (0.1)	100	101
MM87	5 × 10^6^ (0.5)	60	31
	1 × 10^7^ (0.5)	100	27
MM58	5 × 10^6^ (0.5)	100	23
MM410	5 × 10^6^ (0.5)	80	72
MM380	1 × 10^7^ (0.5)	30	n.a.
MM96 **	1 × 10^6^ (0.1)	70	114
	5 × 10^6^ (0.1)	80	114

Recipients used in these studies were female FVB/NCrl mice except * male FVB/NCrl mice. MMe cells were injected i.p. except ** s.c. injection. n.a.: euthanization was scheduled before animals reached the end point due to low tumor take rate.

## Data Availability

The data presented in this study are available on request from the corresponding author. Requests for cell lines should be addressed to J.R.T. directly.

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
