# Peer review of "Mesothelioma Mouse Models with Mixed Genomic States of Chromosome and Microsatellite Instability"

_cancers, 2022, doi:10.3390/cancers14133108_

Round 1
Reviewer 1 Report
Very interesting design and conclusions with potential to attract and facilitate further studies.
Author Response
Thank you for your comments!
Reviewer 2 Report
This study uses the well-established classical tumorigenic approach to characterize and study the tumor inducing capacity 6 mouse mesothelioma cell lines. The Aim of the authors is to determine if their model could be used to predict response to therapy.
There are 2 critical issues that should be addressed in more detail: first the authors should provide more definitive evidence that all 6 cell lines are in fact mesothelioma cell lines. Secondly, the authors should conduct more precise CIN analyses comparing genetic damage before and after inoculation in mice and among different tumor-derived cell lines from tumor nodules induced by the same cell line. If the genetic damage remains stable, then these cell lines could be used to predict response to therapy, if instead there is a high rate of chromosomal instability –as I would predict based on their data- and the genetic make up of the cells changes significantly upon injection in mice, and thus all sub-clones are markedly different, then these cells cannot be used to predict response to therapy.
More specifically,
Lines 252-9: Characterization of the 6 cell lines used to confirm their mesothelial origin:
Although absence of mesothelin argues against mesothelioma, Mesothelin expression is non-specific and cannot be used as a marker of mesothelioma cell lines. Mesothelin is expressed on ~50% of all tumor types. Accordingly mesothelin is not used to diagnoses human mesothelioma, and there is no evidence that mesothelin expression is more specific for mesothelioma in mice. Transmission electron microscopy and WT1 expression, possibly together with additional IHC markers and appropriate positive and negative controls should be used instead to verify the tissue origin of these 6 cell lines.
Lines 322-340: CIN
Given the marked chromosomal instability of all 6 cell lines it would be important to compare each of these cell lines before they were injected into mice, to the tumor derived cell lines from these mice. The results of all 6 cell lines should be shown, before and after inoculation into mice. Ideally at least two derived cell lines should be tested per each injected cell line to address the issue of chromosomal stability in these lines and variability among different tumors from the same cell. It seems quite likely that the authors will find much variability in SKY analyses before and after injection into mice and also among different tumor nodules induced by the same cell line. This evidence would underscore the difficulty of making any prediction about therapy based on data on a single cell line at a given data point. Conversely if the cell lines are stable then they could be used to predict response to therapy
Author Response
This study uses the well-established classical tumorigenic approach to characterize and study the tumor inducing capacity 6 mouse mesothelioma cell lines. The Aim of the authors is to determine if their model could be used to predict response to therapy.
There are 2 critical issues that should be addressed in more detail: first the authors should provide more definitive evidence that all 6 cell lines are in fact mesothelioma cell lines.
The 6 cell lines were indeed derived from mouse mesotheliomas that were characterized by a highly experienced experimental pathologist (Andres Klein Szanto, M.D., Ph.D.), who diagnosed the 6 malignant pleural mesotheliomas (MPMs) based on H&E histopathology staining as well as by IHC with a panel of immunohistochemical markers routinely used to collectively assist in the diagnosis of MPM, including WT1, cytokeratin, and mesothelin. He also used calretinin, although that stain did not always work well on mouse tissues. The cell lines were isolated from ascitic fluid and/or peritoneal lavage as described in the manuscript. Primary cell cultures were used for molecular analyses at passages ≤6. PCR analysis was carried out to ensure that the cell cultures expressed mesothelial markers, and immunoblotting was performed on some of the cultures to validate the PCR results.
Reverse transcription-PCR (RT-PCR) was used to evaluate the cell lines for expression of WT1, cytokeratin 18 (CK18), cytokeratin 19 (CK19), mesothelin, E-cadherin, and N-cadherin. WT1 expression in these cell lines was further confirmed by IHC staining (data added at the revision; Figure S2). PCR primers were used to confirm the presence or knockout of Nf2, p16Ink4a exon 1a, and p19(Arf) exon 1β (Altomare et al., Cancer Res. 2005, PMID: 16166281; Altomare et al., PLoS One. 2011, PMID: 21526190; Menges et al., Cancer Res. 2014, PMID: 24371224). Note that losses of Nf2, p16Ink4a, and p19Arf are hallmark genetic changes in mesothelioma cells, being among the most frequent genetic alterations in this disease. Moreover, the cell lines were derived from asbestos-exposed Nf2+/- (one cell line) or Nf2+/-;Cdkn2a+/- mice that developed mesotheliomas with biallelic inactivation of Nf2, p16Ink4a, and p19Arf (note: the latter two genes are alternate reading frames of the Cdkn2a locus). Thus, these cell lines have both the mesothelioma markers and genetic changes that are expected in bona fide malignant mesothelioma cells.
Secondly, the authors should conduct more precise CIN analyses comparing genetic damage before and after inoculation in mice and among different tumor-derived cell lines from tumor nodules induced by the same cell line. If the genetic damage remains stable, then these cell lines could be used to predict response to therapy, if instead there is a high rate of chromosomal instability –as I would predict based on their data- and the genetic makeup of the cells changes significantly upon injection in mice, and thus all sub-clones are markedly different, then these cells cannot be used to predict response to therapy.
Thank you for reviewer’s valuable suggestions. It is beyond the scope of this paper to perform more CIN analyses by comparing genomic stability among cell lines, tumors, and new cell lines derived from tumor nodules induced by the same cell line. This type of analysis has been considered in our future characterization of these lines. There is no doubt that a therapy predictive model will have great value. However, it was not our aim to report such a model in this paper. We have observed heterogeneity of chromosome instability among the cell lines and cells from the same cell line. Thus, it is expected that the tumors generated by injecting these cell lines into mice will continue to evolve in vivo and will show inter- and intra-tumor CIN heterogeneity, which would recapitulate the situation in many human tumors including mesothelioma. Thus, it would be ideal to test a therapy in such a cell line model in the future.
More specifically,
Lines 252-9: Characterization of the 6 cell lines used to confirm their mesothelial origin:
Although absence of mesothelin argues against mesothelioma, Mesothelin expression is non-specific and cannot be used as a marker of mesothelioma cell lines. Mesothelin is expressed on ~50% of all tumor types. Accordingly mesothelin is not used to diagnoses human mesothelioma, and there is no evidence that mesothelin expression is more specific for mesothelioma in mice. Transmission electron microscopy and WT1 expression, possibly together with additional IHC markers and appropriate positive and negative controls should be used instead to verify the tissue origin of these 6 cell lines.
The authors agree with the reviewer that mesothelin is not a specific marker for mesothelioma. In addition to mesothelin, we have assessed other markers (WT1, CK18, CK19, E-cadherin, and N-cadherin) in these cell lines using RT-PCR. WT1 expression was further confirmed by IHC staining (data added at the revision; Figure S2). Please see the response to reviewer’s 1st question. In addition, tumors from animals injected with these cell lines had strong WT1 expression by IHC (Figure 3b), which indirectly demonstrate that these cell lines are mesothelial in origin.
Lines 322-340: CIN
Given the marked chromosomal instability of all 6 cell lines it would be important to compare each of these cell lines before they were injected into mice, to the tumor derived cell lines from these mice. The results of all 6 cell lines should be shown, before and after inoculation into mice. Ideally at least two derived cell lines should be tested per each injected cell line to address the issue of chromosomal stability in these lines and variability among different tumors from the same cell. It seems quite likely that the authors will find much variability in SKY analyses before and after injection into mice and also among different tumor nodules induced by the same cell line. This evidence would underscore the difficulty of making any prediction about therapy based on data on a single cell line at a given data point. Conversely if the cell lines are stable then they could be used to predict response to therapy
The authors wish to thank the reviewer for valuable suggestions. It is beyond the scope of this paper to perform more CIN analyses by comparing genomic stability among cell lines, tumors, and new cell lines derived from tumor nodules induced by the same cell line. This type of analysis has been considered in our future characterization of these lines. As stated in the response to reviewer’s 2nd question, it was not our aim to use the model to predict a therapeutic response in this paper. As we know, continual evolvement of tumor cells in vivo is one of the characteristics of human cancers. Thus, it won’t be surprising if there are any differences in chromosome instability in tumors derived from these cell lines. However, we expect that some early changes (e.g., driver mutations) will be conserved among all or most subclones, and the CIN differences between tumors/new cell lines and original cell lines may consist of variations observed in these original cell lines, such as multiple copies of whole chromosomes.
Reviewer 3 Report
The authors have submitted a MS to show how they have established a syngeneic orthotopic model of murine Malignant Mesothelioma (MMe) cells derived from an asbestos-induced Cdkn2a+/-;Nf2+/- MMe mouse model. These cells showed a mixed state of CIN and MSI. The authors conclude that the model proposed could be used to test combined immunotherapies and/or developing new therapies for this stubborn neoplasm
One can certainly agree on the proposed syngeneic model however a few more details will provide a more complete scenario to the readers:
The authors should include and discuss the results of the first paper pioneering genomic instability as a marker of asbestos exposure (Cancer Res. 2002 Oct 1;62(19):5418-9. PMID: 12359747
The role of ICi for MMe is certainly attracting attention but recently severely assessed in three papers that cast more than a shadow on their use and should be mentioned and discussed: JAMA Netw Open. 2022 Mar 1;5(3):e221490. doi: 10.1001/jamanetworkopen.2022.1490. PMID: 35262715; PMCID: PMC8908075; JTO Clin Res Rep. 2022 Jan 12;3(3):100280. doi: 10.1016/j.jtocrr.2022.100280. PMID: 35243411; PMCID: PMC8861643; J Chemother. 2022 Apr 12:1-5. doi: 10.1080/1120009X.2022.2061183. Epub ahead of print. PMID: 35411826.
Immune response to MMe cells is hampered by a complex network and the effects of other factors other than CIN and MSI should be included and discussed Int J Cancer. 1998 Dec 9;78(6):740-9. doi: 10.1002/(sici)1097-0215(19981209)78:6<740::aid-ijc12>3.0.co;2-5. PMID: 9833768.; Int J Mol Med. 2003 Feb;11(2):161-7. PMID: 12525871.
A very minor comment is that the abbreviation MM is more often for Multiple Myeloma whereas MPM or MMe is a more suitable abbreviation for Malignant Pleural Mesothelioma or Malignant Mesothelioma respectively
Author Response
The authors have submitted a MS to show how they have established a syngeneic orthotopic model of murine Malignant Mesothelioma (MMe) cells derived from an asbestos-induced Cdkn2a+/-;Nf2+/- MMe mouse model. These cells showed a mixed state of CIN and MSI. The authors conclude that the model proposed could be used to test combined immunotherapies and/or developing new therapies for this stubborn neoplasm
One can certainly agree on the proposed syngeneic model however a few more details will provide a more complete scenario to the readers:
The authors should include and discuss the results of the first paper pioneering genomic instability as a marker of asbestos exposure (Cancer Res. 2002 Oct 1;62(19):5418-9. PMID: 12359747.
We have included the paper mentioned in the revised Discussion (lines 446 - 449; please see below).
“Consistently, high frequency of micronuclei containing whole chromosomes or damaged chromosome fragments has been observed in peripheral blood lymphocytes in pleural MMe, although there was no association between micronuclei and asbestos exposure [64].”
The role of ICi for MMe is certainly attracting attention but recently severely assessed in three papers that cast more than a shadow on their use and should be mentioned and discussed: JAMA Network Open. 2022 Mar 1;5(3):e221490. doi: 10.1001/jamanetworkopen.2022.1490. PMID: 35262715; PMCID: PMC8908075; JTO Clin Res Rep. 2022 Jan 12;3(3):100280. doi: 10.1016/j.jtocrr.2022.100280. PMID: 35243411; PMCID: PMC8861643; J Chemother. 2022 Apr 12:1-5. doi: 10.1080/1120009X.2022.2061183. Epub ahead of print. PMID: 35411826.
As suggested, we have included these papers in the revised Discussion (lines 490 - 500; please see below).
“Analysis of different treatments from different clinical trials for MMe (e.g., immunotherapy versus platinum-based chemotherapy) have been performed and reported recently. Messori et al. found that nivolumab plus ipilimumab or pembrolizumab monotherapy showed a small but significant survival benefit, but not durvalumab plus pemetrexed plus cisplatin, compared to standard of care treatment (pemetrexed plus cisplatin) [96]. However, Meirson et al. [97] and Kerrigan et al. [98] reported no survival benefit for nivolumab plus ipilimumab or either single-agent alone compared to bevacizumab plus cisplatin plus pemetrexed or single-agent chemotherapy. MMe patients in these trials were not stratified by their MSI status or tumor mutation burden. Thus, it remains to be determined whether MMe patients with MSI-H would respond better to immunotherapy than MSS MMe patients.”
Immune response to MMe cells is hampered by a complex network and the effects of other factors other than CIN and MSI should be included and discussed Int J Cancer. 1998 Dec 9;78(6):740-9. doi: 10.1002/(sici)1097-0215(19981209)78:6<740::aid-ijc12>3.0.co;2-5. PMID: 9833768.; Int J Mol Med. 2003 Feb;11(2):161-7. PMID: 12525871.
We have included these papers in the revised Discussion (lines 500 - 508; please see below).
“Moreover, immune response can be hampered by a complex network and many factors other than CIN and MSI. It has been shown that expression of surface molecules (e.g., MHC class II, ICAM-1 and B7-2) in MMe cells is required for antigen presentation [99], and IFN-γ production in an anti-PPD (purified protein derivative) CD4+ T-cell clone was inhibited by TGF-β released by PPD-presenting MMe cells [100]. Thus, appropriate mouse models are needed for testing the effects of these factors on immune response to different therapies. The model we established here will be suitable to address whether the mixed MSI and CIN state will alter the response to immunotherapy.”
A very minor comment is that the abbreviation MM is more often for Multiple Myeloma whereas MPM or MMe is a more suitable abbreviation for Malignant Pleural Mesothelioma or Malignant Mesothelioma respectively.
We have changed the abbreviation MM to MMe, as suggested by the reviewer.
Round 2
Reviewer 2 Report
The authors revised some parts of the paper, but unfortunately they did not conduct further analysis on the cell lines as suggested by the reviewer. The explanation from the authors is that "...it was not our aim to use the model to predict a therapeutic response in this paper. "
This is very puzzling and it raised even more concern, as the authors stated very clearly in many places in the paper that the model they established is suitable for testing preventive and therapeutic interventions. In fact, even in the abstract, they wrote that "This orthotopic model with rapid development of MMe without asbestos exposure is suitable for testing interventions for MMe prevention and therapy..." Instead, if as the authors stated in the response to the reviewer's question that it is not their aim to use this model to predict a therapeutic response, what is the purpose of generating this model then? And what this model can be used for? I do not think the authors have addressed the questions thoroughly. Instead, it indeed raised more concerns.
Author Response
We apologize that the objective of our paper has not been clear to the reviewer. For clarification, our aim was to characterize in detail cell lines from asbestos-induced mesotheliomas derived from genetically engineered mice exhibiting deletions of two of the key tumor suppressor genes that have been implicated in the human disease counterpart. We report that these cell lines rapidly form tumors in syngeneic mice with high penetrance and show a mixed state of CIN and MSI, as in the human disease. The reviewer questioned whether it is our intention to use these cell lines to predict a therapeutic response. Indeed, in addition to studying mesothelioma biology in greater detail in vivo, of course the long-term goal we have for this model system is to be able to use it to predict therapeutic response to novel combined immunotherapies and/or new therapies for this uniformly deadly malignancy. As suggested by the editors, we now discuss this more specifically in the Discussion section of our revised paper (lines 558-564). We have also revised the text related to the aim/goal of this study (lines 26-27, 46-49, and 114-117). We also discuss both the advantages and the limitations/challenges of our model (lines 547-571). We point out in the Discussion that a key intended use of our model is to represent genomic instability and specific molecular targets (e.g., Nf2/hippo, p16Ink4a/CDK4/6/Rb, and DNA repair pathways, as well as high expression of TERT and MSLN) for therapeutic or preventive interventions to enable preclinical proof of concept for the intervention. Development of this information can be used to build a case for clinical testing of the intervention.
Round 3
Reviewer 2 Report
The authors revised the manuscript and discussed further regarding the research purpose of this study and the future goals. It is acceptable for publication.